# Advances in the Production of Sustainable Bacterial Nanocellulose from Banana Leaves

**DOI:** 10.3390/polym16081157

**Published:** 2024-04-20

**Authors:** David Dáger-López, Óscar Chenché, Rayner Ricaurte-Párraga, Pablo Núñez-Rodríguez, Joaquin Morán Bajaña, Manuel Fiallos-Cárdenas

**Affiliations:** 1Facultad de Ciencias e Ingeniería, Universidad Estatal de Milagro, Milagro 091050, Ecuador; ddagerl@unemi.edu.ec (D.D.-L.); ochenchel@unemi.edu.ec (Ó.C.); rricaurtep@unemi.edu.ec (R.R.-P.); 2Facultad de Ciencias Agrarias, Campus Milagro, Universidad Agraria del Ecuador, Milagro 091050, Ecuador; pnunez@uagraria.edu.ec (P.N.-R.); jmoran@uagraria.edu.ec (J.M.B.); 3Escuela Superior Politécnica del Litoral, ESPOL, Facultad de Ingeniería en Mecánica y Ciencias de la Producción, Campus Gustavo Galindo, Km. 30.5 Vía Perimetral, P.O. Box 09-01-5863, Guayaquil 090902, Ecuador

**Keywords:** bacterial nanocellulose, banana leaves, circular bioeconomy, biopolymer, sustainability, simplex-centroid design, fermentation, optimization, yield, biomass

## Abstract

Interest in bacterial nanocellulose (BNC) has grown due to its purity, mechanical properties, and biological compatibility. To address the need for alternative carbon sources in the industrial production of BNC, this study focuses on banana leaves, discarded during harvesting, as a valuable source. Banana midrib juice, rich in nutrients and reducing sugars, is identified as a potential carbon source. An optimal culture medium was designed using a simplex-centroid mixing design and evaluated in a 10 L bioreactor. Techniques such as Fourier transform infrared spectroscopy (FTIR), scanning electron microscopy (SEM), X-ray diffraction (XRD), and thermogravimetric analysis (TGA) were used to characterize the structural, thermal, and morphological properties of BNC. Banana midrib juice exhibited specific properties, such as pH (5.64), reducing sugars (15.97 g/L), Trolox (45.07 µM), °Brix (4.00), and antioxidant activity (71% DPPH). The model achieved a 99.97% R-adjusted yield of 6.82 g BNC/L. Physicochemical analyses revealed distinctive attributes associated with BNC. This approach optimizes BNC production and emphasizes the banana midrib as a circular solution for BNC production, promoting sustainability in banana farming and contributing to the sustainable development goals.

## 1. Introduction

Cellulose is a biopolymer with a high molecular weight, composed of glucose units linked by β-1,4-glycosidic bonds [1]. The molecular formula of cellulose is (C_6_H_10_O_5_)n, where ‘n’ represents the number of linked glucose units that comprise the polymer chain. It serves as the primary component of plant biomass, derived from sources such as plants and wood [2]. Various organisms, including green algae (Valonia) and microorganisms, such as Gram-negative bacterial species like Acetobacter and Agrobacterium [1,3,4,5,6], as well as Gram-positive bacteria like Lactobacillus [5], participate in cellulose synthesis. This particular form of cellulose is referred to as bacterial cellulose (BC) or bacterial nanocellulose (BNC). The Gram-negative bacterium *Gluconacetobacter xylinus*, formerly known as *Acetobacter xylinum*, stands out as the predominant producer of BNC [3,7].

Plant cellulose and BNC exhibit both similarities and differences in structure and properties. Both are celluloses with a similar chemical structure [8]. However, BNC is characterized by higher crystallinity and a greater degree of polymerization compared to plant cellulose, imparting superior strength and elasticity [4,9]. Additionally, the crystalline structure of BNC contains nanostructures that form a network of microfibrils and macrofibrils, resulting in a high water absorption capacity and improved mechanical stability [10]. Furthermore, it is characterized by a more homogeneous nanostructure, primarily composed of Iα and Iβ allomorphs [4,11,12]. The structure of BNC may vary depending on its origin and extraction method. This biopolymer lacks lignin, hemicelluloses, and pectin, which are typically present in cellulose derived from plants [11,12,13,14]. It is noteworthy that the cost of BNC production is typically higher than that of the plant cellulose process [13,15,16].

Cellulose can be produced through two main strategies: chemical synthesis and biochemical synthesis (fermentation). In the chemical synthesis process, it unfolds in two phases. Initially, the isolation and purification of cellulose from lignocellulosic biomass (LB) are performed using mechanical and chemical methods. During this stage, the LB undergoes physical treatments to reduce its size, followed by an intensive chemical treatment involving pulping and bleaching with sodium hydroxide (NaOH) and sodium sulfide (Na_2_S), followed by an additional bleaching stage with chlorine dioxide (ClO_2_) [17]. In the second phase, mechanical homogenization or acid hydrolysis is carried out to produce cellulose nanofibers (CNF) and cellulose nanocrystals (CNC), respectively [18]. The cellulose content can be affected by the pulping or alkaline bleaching process, resulting in a decrease in cellulose yield [19,20,21]. This situation implies the need for the costly recovery of other components present in the lignocellulosic biomass (LB), such as hemicellulose and lignin, found in black liquor pulp [22,23]. Additionally, the production of CNF or CNC presents economic and environmental challenges due to high energy consumption and the use of corrosive mineral acids, leading to the generation of acidic waste [13].

On the other hand, the fermentation or biochemical synthesis methodology is a bottom-up approach. Microorganisms utilize various compounds present in the biomass, such as monosaccharides and nitrogen compounds, to produce diverse products of industrial interest [3,10,24,25]. This process encompasses physical processes [26], as well as chemical and biological treatments to primarily extract sugars from lignocellulosic biomass [8,27,28,29]. Certain bacteria can synthesize BNC by secreting it in the outer membrane of their cells [30,31,32]. Glucan chains are first synthesized in the inner membrane and then secreted to form nano-sized protofibrils (2–4 nm). These protofibrils tightly aggregate to form BNC fibers with a diameter of 20–100 nm [6,13,22]. BNC synthesis is more efficient in the purification and extraction of cellulose compared to vegetable cellulose, as it is produced without lignin and other associated impurities [9,15,32]. BNC has gained popularity due to its simple production process, non-toxicity, high purity, biocompatibility, and environmental friendliness [31,33]. BNC can be obtained from various types of biomasses through sustainable strategies, such as the production of kombucha tea. This beverage is fermented using a symbiotic culture of yeast and bacteria (SCOBY) [34,35]. It can also be obtained in the production of a dessert known as ‘coconut cream’ [36,37].

Figure 1 illustrates the various techniques employed in the production of bacterial cellulose (BC) using different carbon sources. At the biomass level, the following two main categories can be identified: (i) residual biomass (RB), encompassing industrial wastewater and food waste, among others; and (ii) lignocellulosic residual biomass (LRB), primarily composed of rigidly assembled cellulose, hemicellulose, and lignin [38,39,40]. It should be noted that, under certain circumstances, LRB may contain oil, starch, proteins, and juices [41]. These juices may contain reducing sugars (RS) used as a carbon source in fermentation processes [25,26]. In cases where LRB does not contain RS-rich juices, several procedures, such as physical pre-treatment, chemical hydrolysis, or enzymatic hydrolysis, are necessary. The aim of these methods is to expedite the production of cellulose-derived sugars to serve as carbon sources for BNC production [27]. It is crucial to recognize that globally, the availability of waste biomass presents substantial potential for use in sustainable and environmentally friendly production opportunities [3]. In addition to the carbon sources mentioned above, processed substrates such as sucrose or glucose can also be used. Glucose is frequently used as a reagent in laboratory conditions, employing the Hestrin-Schramm (HS) culture technique, which is widely recognized for its efficacy in the production of BNC [42,43]. Processed sugar is commonly used as a carbon source for the production of fermented kombucha tea [44].

BNC stands out as a highly promising natural polymer in scientific research [3]. Its nanostructure and exceptional properties render it a valuable biomaterial with diverse applications [30]. Moreover, BNC exhibits potential as a sustainable alternative to petrochemicals in various industrial applications owing to its remarkable properties [13]. In the biomedical field, researchers have synthesized wound dressing membranes composed of BNC, incorporating antimicrobial plant extracts through the ex situ incorporation technique [45]. Furthermore, investigations have demonstrated that BNC films, enriched with antibiotics such as amoxiclav and fluconazole, effectively serve as antibacterial and antifungal materials, contributing to wound healing [4]. However, the scope of BNC transcends the biomedical realm. Its applicability has been explored in the paper manufacturing industry, where it has been studied as a reinforcing agent [46]. Despite the advantages BNC offers over plant cellulose, its production process incurs significant costs. To address this, research has been directed towards the potential utilization of industrial and agricultural waste, including spent brewer’s yeast, acid by-products from alcohol production, sugar beet molasses, whey, sucrose, acid hydrolysate from potato peel waste, coconut husks, and banana leaves [47]. In an effort to enhance efficiency and reduce production costs associated with BNC, various studies have been conducted [26,48,49,50,51]. 

During banana cultivation, diverse lignocellulosic residues are generated, including the banana leaf (BL), which contains a juice lodged in the midrib and is enriched with nutrients and reducing sugars. Frequently, this residual biomass is left to decompose in agricultural soil, posing potential environmental and health hazards [52]. To date, there remains a substantial lack of characterization regarding the physical and chemical properties of banana midrib juice (BMJ). Nevertheless, studies have demonstrated the efficacy of BMJ in bioethanol production [25] and bacterial cellulose generation [26]. Furthermore, no investigations have yet focused on optimizing the process for obtaining BNC from BMJ.

Experimental design (DoE), particularly in its mixture designs (MDs) modality, constitutes a valuable resource for optimizing procedures and product design in various domains [53,54]. Through the use of DoE, researchers have the ability to explore multiple process parameter configurations with the aim of optimizing product quality and reducing costs [55,56,57,58]. Considering the successes achieved by MDs in improving processing conditions and developing novel products [59,60,61], there is a potential application for their adoption in the manufacturing of various products [56,62]. Based on mixture experiments, some modified mixture design methods have been developed, including the simplex-centroid design method. This design method is commonly applied in the formulation of industrial products such as food processing, chemical formulations, material design, textile fibers, and pharmaceutical drugs [62,63,64]. However, to date, specific information regarding the implementation of the simplex-centroid mixture design in the BNC generation process is lacking.

This study aimed to achieve several objectives: firstly, to analyze the physicochemical properties of both BMJ and the components used in the fermentation medium, such as green tea infusion (GTI) and SCOBY. Subsequently, it aimed to assess the composition of SCOBY through metagenomic techniques. Thirdly, the simplex-centroid mixture design (SCMD) was employed to identify the optimal combination of culture media to maximize the yield in BNC production from BMJ. Lastly, the information gathered from the SCMD was utilized to scale up production to a 10 L reactor, followed by a detailed assessment of the physical and chemical properties of the obtained BNC.

## 2. Materials and Methods

The initial part of this section provides a detailed overview of the procedures employed to analyze the physical and chemical properties of BMJ, GTI, and SCOBY. Following this, the experimental design based on the simplex-centroid methodology and its validation concerning the scalability of production to a 10 L capacity are delineated. Moreover, it includes a description of the methods used to assess the physical and chemical properties of the BNC obtained in the scaled-up medium. 

### 2.1. Materials

Banana leaves were gathered from a conventional plantation of Cavendish subgroup bananas (*Musa acuminata*) situated in Milagro city, Guayas province, Ecuador. The protocol outlined by [26] was followed to extract and process the BMJ. This process involved pasteurizing the BNJ at 121 °C for 15 min. The pasteurized BMJ resulting from the experiment was stored in amber glass jars at a constant temperature of 4 °C for subsequent analysis. The green tea utilized in this research was procured from a local tea processing company and prepared using conventional methods. The GTI was prepared by immersing bagged green tea, employing a ratio of 10 g of green tea leaves per liter, into distilled water heated to 95 °C. The tea steeped for 5 min, after which the bags were removed, and the infusion was kept at 37 °C in a water bath (NTT-20S; Eyela, Tokyo, Japan) until it was employed in formulating various fermentation media and subsequent analyses. SCOBY originated in the microbiology laboratory at the State University of Milagro. It encompasses strains of *Komagataeibacter hansenii* ATCC 23769, *Brettanomyces bruxellensis*, and *Brettanomyces anomalus*.

### 2.2. Analysis of the Physical and Chemical Properties of Banana Midrib Juice, Green Tea Infusion, and SCOBY

The analytical procedures conducted in the BMJ following pasteurization, GTI, and SCOBY are detailed below. The density of the solutions was determined using a pycnometer at 25 °C. Acidity measurements were performed using a digital pH meter (APERA Instruments, LLC-PC60, USA, Columbus, OH, USA), previously calibrated to pH values of 4.0 and 7.0. The concentration of total soluble solids (°Brix) in the JNB at 25 ± 2 °C was assessed using a digital refractometer (Atago Co., Ltd., Tokyo, Japan). To analyze reducing sugars, the 3,5-dinitrosalicylic acid method was employed, quantifying the concentration of reducing sugars using the standard curve of D-glucose [65]. Electrical conductivity was determined using a portable multimeter (HACH-HQ40D, Loveland, CO, USA). 

The chromatic profile was assessed using the CIELAB system, involving the measurement of three parameters: L*, a*, and b*. The L* parameter reflects luminosity (ranging from 0 for black to 100 for white), while a* and b* indicate chromatic variations between green/red and blue/yellow hues, respectively. A Konica Minolta CR5 benchtop colorimeter (Tokyo, Japan) was employed for these measurements. 

To assess the antioxidant capacity, the DPPH radical scavenging assay was performed with minor adaptations [66]. A mixture of 100 μL of the sample and 1.9 mL of a DPPH radical solution diluted in methanol in a 1:1 ratio was prepared. After 30 min at room temperature and in darkness (23–26 °C), the free radical scavenging capacity was evaluated by measuring the absorbance at 517 nm. Results were expressed as mean percentages of inhibition and were conducted in triplicate to ensure data reliability.

### 2.3. Experimental Design for Bacterial Nanocellulose Production

The response surface methodology employed a simplex-centroid mixing experimental design to formulate the medium. This design considers 2*^q^* − 1 combinations of mixtures, where q represents the number of components of the system, namely the BMJ, GTI, and SCOBY. The factors represent the fraction of each raw material in the mixture, ranging from 0 to 1, with no restrictions on the design space.

The feasible space for a three-component mixture experiment is defined as a simplex, which is a triangle. The composition of each mixture varies based on its position in this region. The vertices of the simplex represent pure mixtures, which are composed of 100% of a single ingredient [45,50].

The mathematical model used is as follows:(1)$$Y=\sum_{1\leq i\leq q}^{q}\beta_{i}x_{i}+\sum_{1\leq i\leq j\leq q}^{q}\beta_{i}x_{i}x_{j}+\sum \beta_{123}x_{1}x_{2}x_{3}$$
where *Y* represents the dependent variable, bacterial nanocellulose yield during the 14-day fermentation period; *x*_1_, *x*_2_, and *x*_3_ correspond to the independent variables BMJ, GTI, and SCOBY, respectively. Model adequacy was evaluated by examining the R^2^, R^2^-adjusted, and *p*-value. The modeling and optimization were carried out using the MIXEXP library in RStudio software (version 4.0.3).

#### Culture Medium Optimization

During the initial stage, experiments were conducted in 500 mL wide-mouth glass bottles containing 100 mL of culture medium under aerobic conditions. To maintain optimal conditions, the flasks were covered with sterile paper to prevent light exposure and incubated at a constant temperature of 30 °C for 14 days. At the end of this period, BNC films were collected from the surface of the culture medium. The collected films underwent a thorough washing process to eliminate bacterial cells using a 0.4 N NaOH solution at 80 °C, followed by rinsing with distilled water until achieving a neutral pH. All experiments were conducted in triplicate to ensure result reproducibility. The purified BNC films were freeze-dried at −80 °C for 24 h and weighed (*Wd*). The study analyzed the yields of BNC film production, calculated using Equation (2) provided by Ref. [67].
(2)$$CN\;production\;yield\;\left(\%\right)=\frac{Wd}{V\left(So-Sf\right)}\times 100$$
where *Wd* is the mass of BNC produced (g), *V* is the reaction volume (L), *So* is the initial substrate concentration in the medium (g/L), and *Sf* is the final substrate concentration in the medium (g/L).

### 2.4. Production of Bacterial Nanocellulose Using Optimal Values

The optimal values obtained were utilized to scale up the production process to 10 L of medium under static conditions at 30 °C. The pH, turbidity, and BNC yield on a dry basis were assessed at various time points (0, 4, 7, and 14 days).

Additionally, the physicochemical properties of the NCB were evaluated using techniques such as scanning electron microscopy (SEM), Fourier transform infrared spectroscopy (FTIR), thermogravimetric analysis (TGA), and X-ray diffraction (XRD). 

#### 2.4.1. Scanning Electron Microscopy

Following the fermentation phase, the BNC underwent treatment with NaOH and distilled water, followed by a drying process through freeze-drying at −80 °C using the 4.5 L FreeZone^®^ system (Labconco, Kansas City, MO, USA). Subsequently, morphological analysis of the BNC was conducted using an FEI^®^ Inspect S SEM scanning electron microscope (FEI Company, Hillsboro, OR, USA). During the preparation of the SEM images, the dried BNC was fixed and coated with a thin layer of gold nanoparticles. The experiments were performed at a magnification of 10,000× and an accelerating voltage of 14.5 kV.

#### 2.4.2. Fourier Transform Infrared Spectroscopy

The chemical structure of the dry BNC was determined using FTIR. Spectra were recorded using the attenuated total reflectance (ATR) technique on a Spectrum GX spectrometer (PerkinElmer, Waltham, MA, USA) in the range of 4000 to 500 cm^−1^. A total of 32 spectra were accumulated with a resolution of 4 cm^−1^. The spectral data were normalized and subjected to background and baseline corrections. Subsequently, graphical representations were generated using OriginPro 9.0 software (OriginLab Corporation, Northampton, MA, USA).

#### 2.4.3. Thermogravimetric Analysis

The thermal stability of BNC was determined by conducting TGA using a TA^®^ Instruments Q-600 with a simultaneous DTA thermal analyzer. The BNC samples were weighed and placed in a tray with a weight range of 2 to 5 mg. The samples were heated with nitrogen from 25 °C to 1000 °C at a constant heating rate of 10 °C/min.

#### 2.4.4. X-ray Diffraction

The dried BNC was ground to a powder for X-ray diffraction analysis. The sample was analyzed with nickel-filtered Cu-Kα radiation (α = 0.15418 nm). Diffraction maps were recorded at room temperature using a PANalytical^®^ X-ray diffractometer X’Pert, operating at 45 kV and 40 mA. The 2θ range was from 5 to 80°, with a scanning speed of 0.01°/s. The crystallinity index (*CrI*) of the BCs was determined using Segal’s technique [68], as shown in Equation (3):(3)$$CrI\;\left(\%\right)=\left(I_{200}-I_{am}\right)/\left(I_{200}\right)\times 100\%$$

Here, *CrI* represents the crystallinity index, *I_am_* represents the minimum diffraction intensity at around 2θ = 18°, and *I*_200_ is the maximum diffraction intensity of (002) lattice diffraction at around 2θ = 22.6° [69].

## 3. Results and Discussion

### 3.1. Physical and Chemical Characteristics of the Components Utilized in the Preparation of the Culture Medium

In the context of this research, a meticulous analysis of the physicochemical properties inherent to the components of the culture medium formulation has been carried out. The results derived from these evaluations can be found in Table 1.

pH plays a pivotal role in fermentation processes. BMJ exhibited a less acidic nature in comparison to GTI and SCOBY. Nonetheless, Ref. [25] reported a pH of 6.8 for BMJ, whereas Ref. [70] observed a pH of 6.4 ± 0.15 for banana leaf biomass. The observed variability in the pH of JNB may be attributed to factors such as soil type, genetic variability, and the application of chemical fertilizers [67,71]. On the other hand, the pH value of GTI closely aligns with findings from previous studies [72,73,74]. The acidity of GTI may be attributed to the extraction of compounds inherent in tea leaves, including organic acids, phenols, carboxylic groups, and amino acids [74]. The pH of SCOBY exhibits analogous values to those documented in other studies [75,76]. It has been postulated that maintaining an acidic pH range, typically ranging from 6 to 3, may foster the proliferation of a broader spectrum of microorganisms, potentially enhancing bacterial nanocellulose production [26].

The determination of total soluble solids (TSS) concentration can be achieved through density and Brix measurements. The TSS values observed in GTI align consistently with those reported in prior investigations [74]. As posited by [73], the presence of phenolic compounds, alkaloids, carbohydrates, amino acids, pigments, vitamins, and other minor compounds contributes to the TSS in GTI. SCOBY exhibits a higher density and °Brix, indicative of a heightened concentration of soluble compounds compared to BMJ and GTI. This elevated concentration may potentially provide a richer nutrient milieu for microbial activity during the fermentation process. 

In comparison to GTI and SCOBY, BMJ had the highest amount of reducing sugars. According to [77], the banana cultivars Williams and Grand Nain had a total soluble carbohydrate content of 25.21 ± 2.84 and 23.50 ± 0.86 mg/g, respectively. Similarly, Ref. [25] found that the pressed juice of banana leaves contained a total sugar of 14%, with glucose (18.9 g/L), sucrose (13.29 g/L), and fructose (15.63 g/L) present. It should be noted that banana composition can be significantly influenced by factors such as soil type and genetic variability [71,77]. 

The electrical conductivity of BMJ suggests a remarkable presence of ions in this solution. Ref. [70] determined that the conductivity of banana leaf biomass was 1.98 ± 0.02 mS/cm, a higher value than reported in this research. According to [25], the juice obtained by pressing banana leaves possesses a significant amount of macroelements (K, Na, and Mg) and traces of microelements (Fe, Cu, Zn, and B). Regarding GTI, Ref. [72] determined three major elements (Ca, K, and Mg) and eleven trace elements (Al, As, B, Cd, Co, Cr, Fe, Mn, Pb, Se, and Zn) in green tea leaves. The findings suggest the absence of significant differences in electrical conductivity between BMJ and GTI, possibly attributable to similarities in ionic composition, indicating uniformity in electrical properties. Ref. [78] indicates that the presence of these macroelements (K, Na, and Mg) and microelements (Fe, Cu, Zn, and B) in the culture medium is considered crucial in microbial fermentation, indicating that they have a significant effect on yeast growth rate, substrate consumption, and fermented beverage production yield. 

On the other hand, BMJ exhibits a remarkable antioxidant capacity of 71.0%, along with a significant concentration of Trolox, highlighting its efficacy in protecting against oxidative damage. Phytochemicals, such as flavonoids, tannins, and terpenoids, are present in banana leaves [79]. Similarly, GTI also demonstrates high antioxidant activity, attributed to the phytochemicals found in green tea. Ref. [74] reported comparable values of 76.71–95.48% inhibition. This capability may provide protection against oxidative damage during fermentation. The results revealed no significant difference in Trolox concentration (µM) between BMJ and GTI. This finding could be attributed to the possible similarity in antioxidant composition. 

The colorimetric parameters L*, a*, and b* offer detailed insights into the luminosity, reddish or greenish tones, and yellow or blue hues present in the analyzed materials. In the case of BMJ, the luminosity value indicates a relatively subdued light shade. The values of a* and b* suggest a minimal tendency towards reddish tones and a slight inclination towards blue tones. In contrast, GTI has a higher L* value, indicating greater brightness or clarity compared to BMJ. However, the a* and b* values suggest a tendency towards greenish and bluish tones. These results align with previous studies [72,74]. Ref. [72] proposes that a* and b* values in GTI are closely linked to elevated Total Dissolved Solids (TDS) and pH, noting that an increase in TDS can intensify the green tea’s color. It is essential to understand the relationship between color attributes and the chemical composition of the product to comprehend the influence of media components on the final product’s visual characteristics. However, the SCOBY has the highest brightness value, indicating greater visual clarity. Additionally, the a* and b* values suggest a tendency towards reddish and yellowish tones. 

The chroma values, representing color saturation, vary significantly among samples. BMJ exhibits the lowest chroma value, indicating lower color saturation. Meanwhile, GTI and SCOBY show significantly higher values, suggesting greater saturation and vibrancy in the hues present in these samples. Brewing water is a key factor influencing tea brew color. Results from the analysis of L*, a*, b*, chroma (C*), and hue (H*) values of BMJ, GTI, and SCOBY samples indicate significant differences in their color characteristics. The SCOBY sample is the brightest, reddest, and yellowest of the three, while the BMJ sample is the darkest, greenest, and least saturated. Such variations could be related to the presence and concentration of biochemical compounds, such as phytochemicals or antioxidants, influencing their visual characteristics and potentially their functional properties in fermentation and biological production applications.

### 3.2. Optimized Culture Medium Using Simplex-Centroid Design

The study utilized a simplex-centroid design to assess the effect of three variables: BMJ, GTI, and SCOBY inoculation, on the optimal yield of bacterial nanocellulose. Table 2 provides detailed information on the experimental design, which generated nine different combinations.

The response variables were analyzed using analysis of variance (ANOVA) to select a suitable model. ANOVA was chosen based on several measured criteria, with a 95% confidence interval, as presented in Table 3. The use of ANOVA allowed for the analysis of the influence of each combination, with a significance level of 0.05. The RStudio statistical package MIXEXP was used to perform this analysis.

The study found that the BMJ, GTI, and SCOBY components had a significant impact on BNC procurement performance (*p*-value < 0.05). Additionally, the mixing ratios of *x*_1_*x*_3_, *x*_2_*x*_3_, and *x*_1_*x*_2_*x*_3_ also had a significant effect. However, the *x*_1_*x*_2_ combination did not significantly affect the yield of bacterial cellulose production (*p*-value > 0.05). 

A regression analysis was conducted on the mixture variables (BMJ, GTI, and SCOBY), revealing that their relationships could be statistically modeled. The model’s performance was assessed using the R^2^ coefficient. According to existing literature, the coefficient of determination (R^2^) indicates conformity when surpassing 70% and is further interpreted favorably as it approaches 1.00, signifying a superior fit of the model to real data or its overall significance [80]. Additionally, the adjusted R^2^ was employed to evaluate the alignment of experimental results with theoretical outcomes [81]. In this context, *p*-values were less than 0.05, and both coefficients of determination (R^2^ and adjusted R^2^) reached 99.99% and 99.97%, respectively. The F-statistic, registering at 3.879 with a *p*-value of 0.0002578, underscores the overall significance of the model. The residual standard error, standing at 0.1018 with 2 degrees of freedom, emphasizes its close agreement with the obtained data. Furthermore, the test for lack of fit, with a *p*-value greater than 0.05, indicated that the model was sufficient to predict BNC production performance. See Equation (4).
Yield = 4.170*x*_1_ + 3.250*x*_2_ + 4.130*x*_3_ + 4.160*x*_1_ × *x*_2_ + 8.080*x*_1_ × *x*_3_ + 5.720*x*_2_ × *x*_3_ + 32.722*x*_1_ × *x*_2_ × *x*_3_(4)

Statistical analysis shows that all three factors have a significant influence on bacterial nanocellulose production. The regression coefficients for each factor are positive, indicating that an increase in any of them correlates with an increase in bacterial nanocellulose production. The model without an intercept is obtained by removing the constant term from the model with an intercept. When all factors are at zero, the yield of bacterial nanocellulose production is considered to be zero. 

The mixing contour plot within the CSD visually represents the impact of variables on BNC production. It uses color coding to differentiate between different levels of performance. Figure 2 exemplifies this representation, where darker shades of red indicate lower performance, while lighter shades and yellow indicate gradual improvement and areas of higher performance, respectively. It is worth noting that a composition closer to the central point leads to a significant increase in the yield of BNC production, reaching 6.5 g BNC/L. The results demonstrate that the optimal concentration of SCOBY is considerably high, while GTI and BMJ have optimal concentrations in an intermediate range. The analysis of the center point suggests a trend toward convergence to a uniform concentration, potentially influenced by the acidic pH of the SCOBY, promoting the composition of the medium [82]. It is worth noting that BMJ possesses components abundant in sugars and nitrogenous compounds, fostering the growth of microorganisms for the production of bioproducts [25,26].

### 3.3. Kinetics of BNC Production Using Optimal Values

Based on the results obtained from the simplex-centroid mixing design, production was scaled up to a 10 L bioreactor and maintained under stationary conditions at 30 °C and 1 atmosphere. Evaluations of pH, turbidity, and BNC production yield were carried out at different time intervals, specifically at 0, 4, 7, 10, and 14 days. All measurements were carried out in triplicate. See Figure 3.

The results show that the pH of the culture medium decreases over time. The decrease in pH observed in kombucha tea-like fermentation media may indicate metabolic activity, as it is associated with the production of organic acids such as acetic, D-glucoronic, lactic, tartaric, and citric acids. Previous studies have suggested that these acids are generated during the fermentative process [26,83,84].

The turbidity of the optimized culture medium experienced a significant increase in the first seven days of fermentation and remained stable thereafter. During the initial phase, bacteria increase their population using oxygen, generating cellulose and turbidity. After oxygen depletion, bacteria located on the surface produce bacterial cellulose, reaching equilibrium, while those submerged in the fermentation medium become inactive below the surface. The latter can be reactivated for future cultures [83]. Regarding turbidity in kombucha analog beverages, this is associated with microbial growth. *Komagataeibacter*, predominant in the initial culture, induces cellulose production, thus exerting a direct influence on turbidity [85,86].

The formation of BNC manifested as a delicate layer, progressively thickening throughout the fermentation process, discernible from the fourth day onwards. By the 14th day, an average yield of 6.82 g BNC/L on a dry basis was attained, signifying promising outcomes. In contrast, antecedent studies reported a yield of 0.031 g/g at 21 days [26], with others achieving up to 6.4 g BNC/L of dry mass after 21 days [82]. The production of BNC from diverse raw materials, such as acerola juice extract and industrial residues, yielded 2.9 g/L after 12 days of fermentation [87]. Similarly, BNC production in black tea media recorded 3.06 g BNC/L in dry weight [88].

The utilization of banana leaf extract and green tea infusion yielded encouraging results, attributed to the richness of nutrients, vitamins, polyphenols, and phytochemicals inherent in these infusions. This strategic approach alleviated the necessity for additional nitrogen sources, consequently yielding a cost reduction [82]. Notably, this achievement can be attributed to the exploitation of symbiotic relationships among bacteria within the SCOBY. As per [89], individual strains producing BNC demonstrated a yield of 3.0–4.1%, while symbiotic cultivation achieved a BNC production ranging from 6.3% to 10.0%, underscoring a substantial enhancement in efficiency. In this context, the selection of residual biomass as a raw material not only satisfies the technical requisites for BNC production but also ensures sustainability by mitigating competition with food resources. This strategic approach presents auspicious perspectives for the proficient and cost-effective production of BNC with specialized applications.

### 3.4. Physical and Chemical Properties of Bacterial Nanocellulose

Bacterial nanocellulose was harvested from both the HS control medium (HS-BNC) and the optimized medium, which was scaled up to a volume of 10 L (SCD-BNC). Subsequently, the membranes underwent a washing process with 0.4 N NaOH, followed by rinsing with distilled water until the washing liquid achieved a neutral pH. The BNC samples were then subjected to a freeze-drying process and analyzed using SEM, FTIR, TGA, and XRD techniques to explore their physical and chemical composition.

#### 3.4.1. SEM Analysis

SEM was employed to examine the microscopic morphology of the BNC samples. Figure 4A,B illustrate the morphology of BNC produced by both the HS control medium (HS-BNC) and the SCD-optimized medium (SCD-BNC), respectively. The BNC was purified through treatment with NaOH, followed by thorough washing with water to eliminate all impurities and bacterial cells [26]. The microstructure revealed a cross-linking of cellulose fibrils in both BNC samples. The images depict the arrangement of BNCs in a layered network with randomly oriented cellulose microfibrils. In the case of HS-BNC, the diameters of the microfibrils ranged from 23.9 nm to 119.2 nm, with an average of 43.5 nm (Figure 4(C1)), consistent with previous findings [29,89]. Conversely, SCD-BNC had diameters ranging from 28.7 nm to 120.0 nm, with a mean of 58.23 nm (Figure 4(C2)). These results suggest a larger mean fiber diameter in the SCD-BNC medium, a conclusion supported by prior research focused on the evaluation of industrial waste [24,27,90]. The density, size, and arrangement of BNC fibrils depend primarily on the medium composition, viscosity, and activity of BNC-producing bacteria [27,91,92].

#### 3.4.2. FTIR Analysis 

The study used FT-IR analysis to evaluate the structural characteristics of HS-BNC and SCD-BNC membranes produced under different nutritional conditions. The analysis revealed characteristic bands associated with type I cellulose. Figure 5 shows similar FTIR spectra between HS-BNC and SCD-BNC, with a peak in the 3300–3412 cm^−1^ region indicating the -OH stretching vibrations characteristic of type I cellulose [93,94]. Despite the similarity, there is a variation in the peak intensity for SCD-BNC in this region, indicating a modification in the hydrogen bond interaction. This alteration is attributed to both the influence of carbohydrates from the carbon source in the culture medium [87,95] and the presence of adsorbed water [96,97].

The samples also presented other relevant bands, such as those corresponding to the CH stretching of CH_2_ groups around 2893 cm^−1^, specific to type I cellulose [90], and the asymmetric CH_2_ stretching at 2854 cm^−1^ [48], which represents the C-H stretching vibration of the sugar rings [27]. In the case of SCD-BNC, the simultaneous detection of both bands suggests a probable influence of the sugar substrate composition on the fermentation process. Previous research has illustrated how different carbon sources can impact the spectrum of the BNC sample [87,98]. Additionally, prominent bands were observed around 2250 cm^−1^, corresponding to CN stretching [99], which could indicate an association with the presence of carbohydrates and nitrogenous compounds found in the ingredients used to formulate the fermentation medium [87,95,100,101]. Furthermore, an absorption band at 1650.5 cm^−1^, attributed to the presence of the carboxyl functional group (C=O), and a band around 1642 cm^−1^, attributed to H-O-H bending due to water adsorption, were observed [100,102].

The spectra of both samples showed characteristic signals of type I cellulose. Strong bands at 1425 cm^−1^ were observed in the spectrum of cellulose. Strong bands at 1425 cm^−1^ were assigned to CH_2_- symmetric bending; a band at 1160 cm^−1^ indicated C-O-C asymmetric stretching at the β-glycosidic bond [27,87], and a maximum at 1068 cm^−1^ corresponded to C-O- symmetric stretching of the primary alcohol. The differences between SCD-BNC and HS-BNC can be attributed to the carbohydrates and phenolic compounds present in the fermentation medium. These phenolic compounds can combine with cellulose in situ during fermentation, which explains the observed changes [89,103].

#### 3.4.3. TGA Analysis 

The thermal stability of BNCs produced in the HS control medium (HS-BNC) and the SCD-optimized medium (SCD-BNC) was assessed through TGA, the results of which are presented in Figure 6A, and differential thermogravimetric analysis (DTG), illustrated in Figure 6B. Both BNC samples displayed similar curves with three distinct stages of mass loss. Initially, a marginal decrease in weight was observed up to approximately 150 °C, attributed to water evaporation during the initial heat treatment. The thermal degradation process unfolded in two discernible stages, with the second stage occurring between 240 and 425 °C and closely linked to cellulose degradation. This stage involved depolymerization, dehydration, and decomposition of the glucose units, culminating in the formation of a carbonaceous residue [104]. It resulted in a weight loss of approximately 56.56% and 50.02% in HS-BNC and SCD-BNC, respectively. The final stage, reaching 800 °C, showed a weight reduction of 15.3% and 18.6% in HS-BNC and SCD-BNC, respectively. Overall, the sample experienced an aggregate mass loss of 82.09% of its initial weight.

Previous studies have indicated that the mass loss of bacterial cellulose usually ranges from 70% to 95%, depending on the bacterial strain responsible for cellulose production [88,105,106]. The HS sample demonstrated a diminished residual in TGA compared to SCD, attributable to distinctions in the medium’s composition and the structure of the produced BNCs [98]. In the DTG curves, the maximum decomposition temperatures (Tmax) for HS-BNC and SCD-BNC were 352.24 and 355.51 °C, respectively. A slight deviation in the degradation temperature could be ascribed to variations in BNC fibril size, arrangement, and compactness [92]. The DTG results aligned with the XRD findings, as higher crystallinity has been reported to improve the thermal stability of BNC [107]. TGA findings for both culture media revealed similar patterns, consistent with the results of previous studies [26,29,99,108].

#### 3.4.4. XRD Analysis 

BNC is a homogeneous polycrystalline macromolecular compound consisting of an ordered crystalline region and a disordered amorphous region [109]. Its remarkable crystallinity, compared to plant cellulose, serves as a distinctive attribute [29]. Within the ordered crystalline region, various BNC samples exhibit broad and robust peaks at 2θ = 14.5° and 22.7°, values consistent with previous research [24,29,110]. The XRD patterns display three reflections at 14.5, 16.7, and 22.7, respectively. See Figure 7.

Representing typical diffraction peaks of type I cellulose and reflecting the crystalline and amorphous structure of BNC components. These results show similarities with previous studies [111,112]. The *CrI* stands as a crucial parameter for BNC and its derived compounds [113]. According to Segal’s formula, the crystallinity of BNC produced in the SCD-optimized medium reached 83.5%, surpassing that of BNC produced in the HS medium (79.5%). These findings suggest that BNC from the medium optimized using BMJ has the highest cellulose Iα content, while BNC from the HS medium has the lowest cellulose Iα content. Various factors, such as cultivation methods, carbon sources, pH of the medium, stirring speed, temperature, cultivation time, additives, and drying methods, could influence the crystallinity of BNC [114]. The possible relationship with NaOH degradation is also considered, as this degradation process initiates the decomposition of the polymer and facilitates the hydrolysis of the cellulose [86,115].

## 4. Conclusions

This study presents a noteworthy advancement in BNC production using sustainable resources such as BMJ and GTI. Optimization through a simplex-centroid experimental design resulted in the development of a statistically significant model, emphasizing its efficiency in compositions near the central point of the design and underscoring the industrial feasibility of utilizing agricultural by-products.

The research contributes significantly to both practical and theoretical aspects. The scalability of the process and its potential in industrial settings are demonstrated through laboratory-scale and bioreactor-scale production. A robust model has been established from a theoretical perspective, highlighting the key influence of the components of the culture medium and thus significantly contributing to the body of knowledge in this emerging field.

This study aligns with recent developments in sustainable biomaterials, emphasizing unique contributions such as the use of BMJ and GTI. It underscores the importance of utilizing renewable resources and optimizing processes to encourage further discussion on the integration of these approaches across various industrial sectors.

However, this study has inherent limitations, such as not exploring other variables. This suggests promising areas for future research, including the investigation of different biomass sources and the optimization of fermentation conditions on an industrial scale. The results demonstrate significant potential for diverse BNC applications in the industry, making a tangible contribution to both the scientific community and broader sustainable development goals.

## Figures and Tables

**Figure 1 polymers-16-01157-f001:**
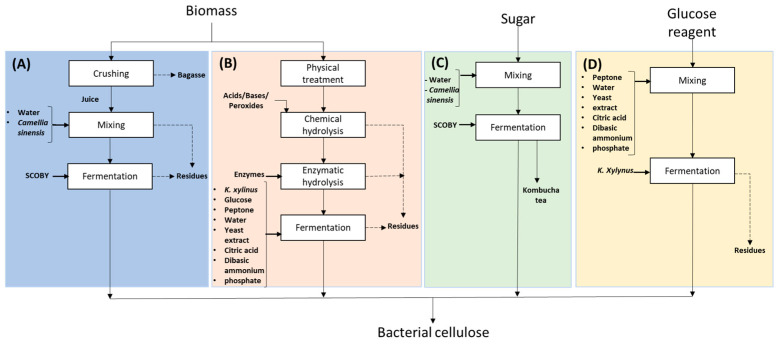
Different approaches for obtaining bacterial cellulose using different carbon sources. (**A**) Residual biomass rich in sugary juices, which requires minimal physical treatments to extract its sugars, which are then used as a carbon source. (**B**) Residual biomass containing sugars requires physical, chemical, or biological pre-treatment to extract the sugars. These sugars are then used as a carbon source in fermentation processes. (**C**) Kombucha tea production uses conventional sugar (sucrose). (**D**) The carbon source in Hestrin-Schramm (HS) medium is reagent-grade glucose.

**Figure 2 polymers-16-01157-f002:**
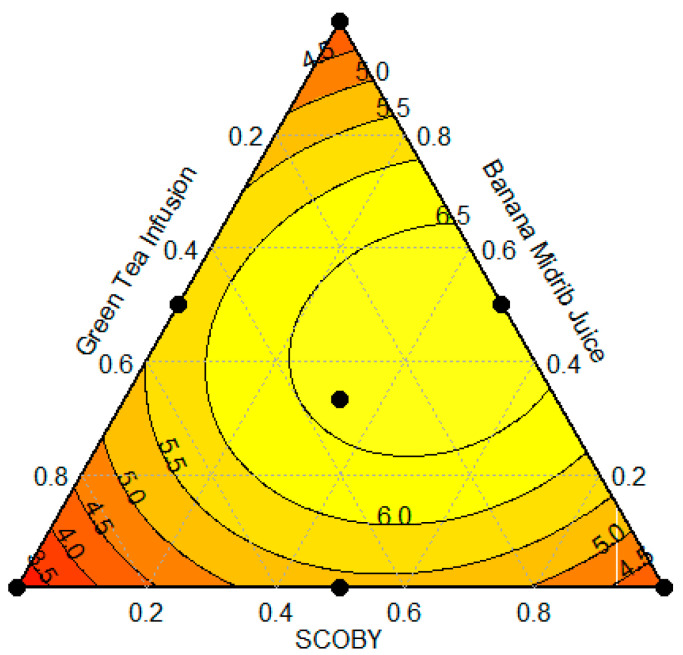
Mixing contour plot showing the effect of the variables. Colour coding makes it possible to distinguish the different yield levels of bacterial nanocellulose production: reddish tones indicate lower yields and yellow tones indicate higher yields.

**Figure 3 polymers-16-01157-f003:**
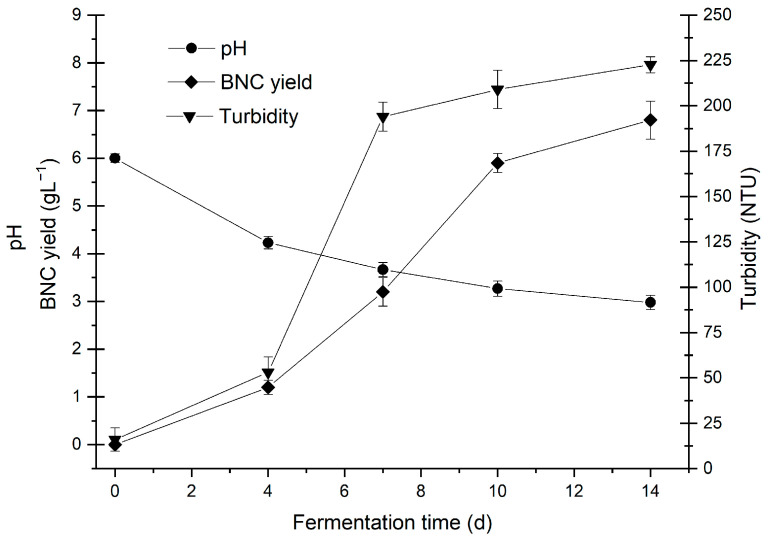
Kinetic analysis of bacterial nanocellulose production: pH and turbidity variations in the culture medium.

**Figure 4 polymers-16-01157-f004:**
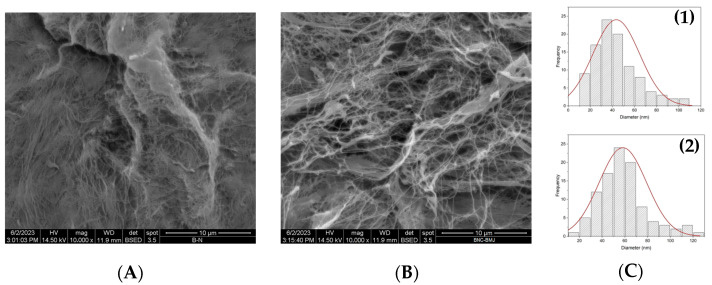
SEM images of the bacterial nanocellulose membranes: (**A**) Obtained in the HS control medium. (**B**) Obtained in the medium optimized by simplex-centroid design (SCD). Panels (**C1**,**C2**) present plots of diameter distribution and standard distribution for HS-BNC and SCD-BNC, respectively.

**Figure 5 polymers-16-01157-f005:**
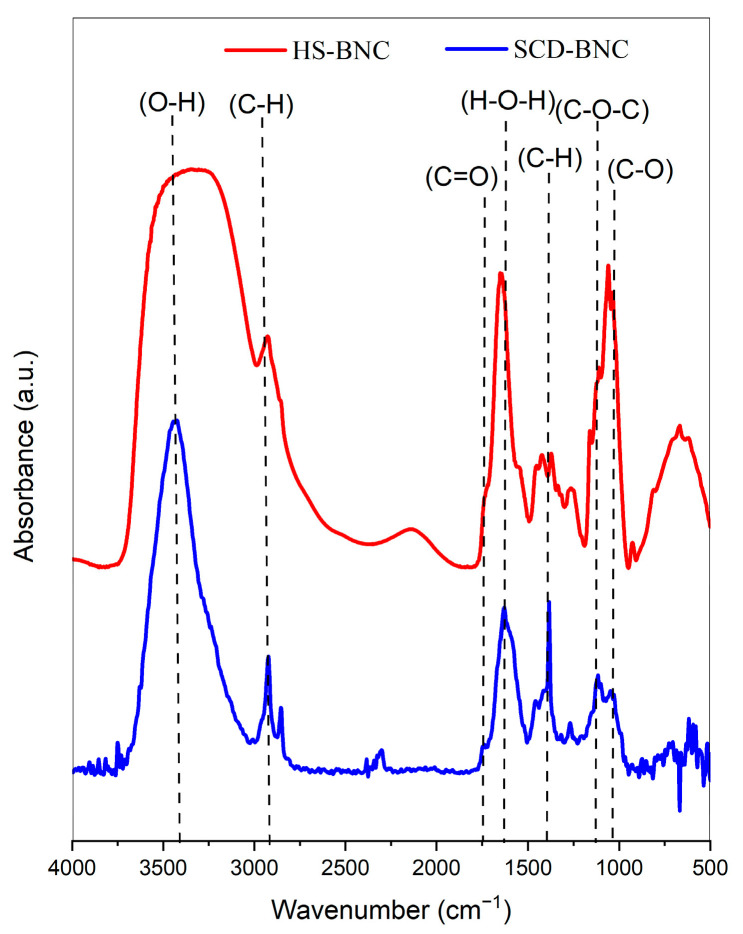
Characterization of bacterial nanocellulose: Fourier transform infrared spectroscopy spectra. The red line represents the BNC spectrum obtained in the HS control medium, whereas the blue line represents the BNC spectrum obtained in the SCD-optimized medium.

**Figure 6 polymers-16-01157-f006:**
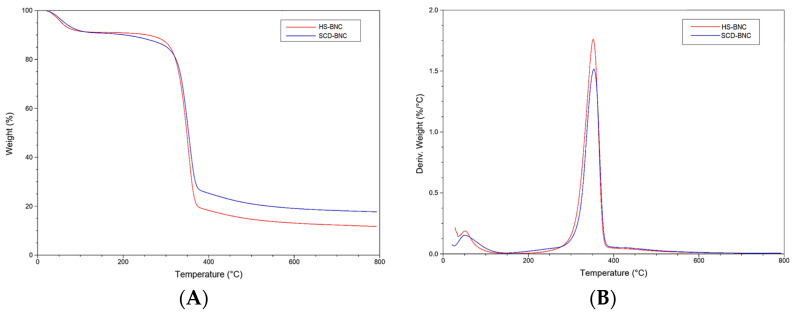
(**A**) Thermogravimetric analysis curve and (**B**) derivative thermogravimetry curve. The red line represents the BNC spectrum obtained in the HS control medium, while the blue line represents the BNC spectrum obtained in the SCD-optimized medium.

**Figure 7 polymers-16-01157-f007:**
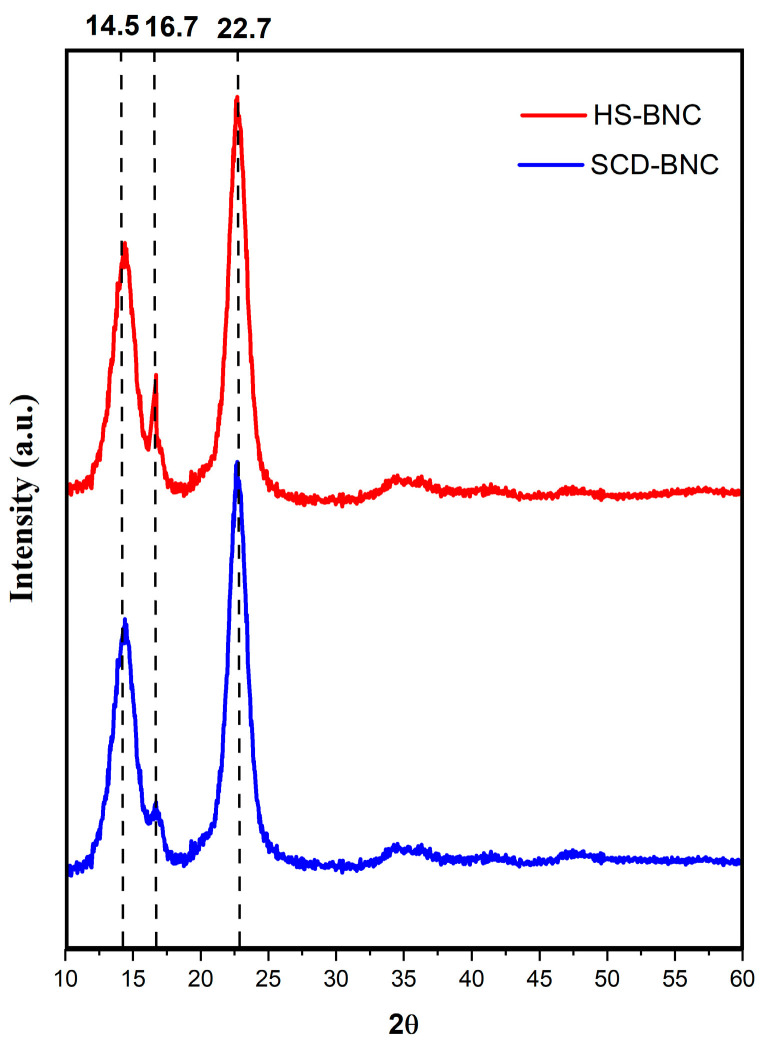
X-ray diffractogram of bacterial nanocellulose in the optimized medium from banana mid-rib juice, SCD-BNC (blue line), and bacterial nanocellulose in the HS medium, BNC (red line).

**Table 1 polymers-16-01157-t001:** Physicochemical properties of individual components in formulation.

Parameters	BMJ	GTI	SCOBY
pH	5.64 ± 0.02	5.37 ± 0.02	3.20 ± 0.02
Density (g/mL)	1.02 ± 5.0 × 10^−3^	1.01 ± 5.0 × 10^−3^	1.05 ± 5.0 × 10^−3^
°Brix	4.000 ± 0.32	1.33 ± 0.32	12.83 ± 0.32
Reducing sugars (g/L)	15.97 ± 0.49	0.00 ± 0.00	5.67 ± 0.49
Conductivity (µS/cm)	16.57 ± 5.70 ^a^	39.40 ± 5.70 ^a^	75.13 ± 5.70
Antioxidant activity (DPPH) (%)	71.00 ± 0.11	70.27 ± 0.11	16.18 ± 0.11
Trolox (µM)	45.07 ± 0.58 ^b^	45.47 ± 0.58 ^b^	8.17 ± 0.58
L*	27.55 ± 0.13	92.23 ± 0.13	90.97 ± 0.13
a*	0.49 ± 0.15	−5.24 ± 0.15	−0.42 ± 0.15
b*	−0.35 ± 0.32	34.48 ± 0.32	15.33 ± 0.32
chroma	0.62 ± 0.16	27.43 ± 0.16	15.13 ± 0.16
Hue	324.03 ± 0.32	210.33 ± 0.32	90.50 ± 0.32

BMJ: banana midrib juice; GTI: green tea infusion; SCOBY: Symbiotic Colony Of Bacteria and Yeast. Means of the three replicates ± standard deviation. The same superscript letters within a row indicate that there are no significant differences (*p* ≤ 0.05 according to the significant differences (*p* ≤ 0.05 according to Tukey’s test).

**Table 2 polymers-16-01157-t002:** Design of a simplex-centroid mixture with three components to obtain bacterial nanocellulose.

Treatment	BMJ	GTI	SCOBY	Yield (g L^−1^)
1	1	0	0	4.17 ± 0.10
2	0	1	0	3.25 ± 0.08
3	0	0	1	4.13 ± 0.12
4	0.5	0.5	0	4.75 ± 0.15
5	0.5	0	0.5	6.17 ± 0.11
6	0	0.5	0.5	5.12 ± 0.09
7	0.33	0.33	0.33	7.13 ± 0.05
8	0.33	0.33	0.33	7.10 ± 0.10
9	0.33	0.33	0.33	6.94 ± 0.13

BMJ: banana midrib juice; GTI: green tea infusion; SCOBY: Symbiotic Colony Of Bacteria and Yeast.

**Table 3 polymers-16-01157-t003:** Regression analysis for bacterial nanocellulose production.

Factors	Coefficient	Std. Error	t-Value	*p*-Value	
*x* _1_	4.170	0.1018	40.973	<0.001	Very Significant
*x* _2_	3.250	0.1018	31.933	<0.001	Very Significant
*x* _3_	4.130	0.1018	40.580	<0.001	Very Significant
*x*_1_:*x*_2_	4.160	0.4986	8.344	<0.05	Low Significant
*x*_1_:*x*_3_	8.080	0.4986	16.206	<0.01	Significant
*x*_2_:*x*_3_	5.720	0.4986	11.472	<0.01	Significant
*x*_1_:*x*_2_:*x*_3_	32.722	2.6968	12.133	<0.01	Significant

## Data Availability

Data are contained within the article.

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
