# Peer review of "Advances in the Production of Sustainable Bacterial Nanocellulose from Banana Leaves"

_polymers, 2024, doi:10.3390/polym16081157_

Round 1

Reviewer 1 Report

Comments and Suggestions for Authors

In general, the article needs improvement in both presentation and results analysis, especially in the characterization techniques of the bacterial cellulose obtained. While there is a good experimental design, the material characterization requires significant enhancements.

Line 50-51: This sentence should be revised, as there are several studies demonstrating the degradability of components such as pectins and hemicelluloses. Lignins are a bit more complex due to the size of their structure, but there are also studies demonstrating their degradability. Please correct the sentence.

Line 60-61: The literature considers that cellulose extracted by different chemical methods seeks not to alter the crystalline structure of cellulose, hence the discussion of cellulose type I with different allomorphs. Please correct this sentence and provide supporting bibliography. 

Line 67: One of the most important factors improving bacterial cellulose production is nitrogen sources. Supplement this sentence with bibliography supporting the importance of substrate components in production efficiencies. 

Line 237: In ATR infrared, the crystal has an effect on the final result. ¿Was any mathematical manipulation of the spectrum performed before comparing it with the literature? 

Line 244: Why was a heating rate of 20°C per minute used, when the international standard in the literature suggests 10°C per minute? How might this affect sensitivity in the assay? 

Line 432: Change cm-1 to cm-1

Line 432: The text presents different ways of citing the bibliography, using numbers in some cases and authors' surnames in others. Please verify this. 

Figure 4b: I recommend adding the DTG and complementing the analysis with what this graph shows. 

Figure 5: The presented diffractogram has several issues. It is recommended to repeat the test or validate if there is a problem in the sample preparation. 

Figure 5: Include the most important angles within the graph. 

Lines 462-471: The X-ray diffraction analysis presents several problems, starting with the location of the amorphous pattern, which does not correspond to what has been studied and is found in the literature. Additionally, the poor definition of the amorphous pattern can ultimately affect the crystallinity calculation. I recommend reading more on the topic and improving the analysis. I recommend reading the following articles:

https://link.springer.com/article/10.1007/s10570-012-9833-y

https://link.springer.com/article/10.1007/s10570-020-03177-8

Lines 475-479: Sample preparation for SEM analysis needs improvement. Dilute them better to analyze the fiber structure obtained under the substrates used. It is recommended to create a graph of diameter distribution and standard deviation of the diameters taken

Comments on the Quality of English Language

The article requires substantial improvements, particularly in the analysis of characterization results. It is highly descriptive and lacks relevant findings compared to existing literature. I recommend not publishing it until significant improvements are made in the results analysis.

Author Response

We sincerely appreciate the time you've taken to review this manuscript. Below, you'll find detailed responses along with the corresponding revisions and highlighted corrections.

Reviewer 2 Report

Comments and Suggestions for Authors

In this work, the authors evaluated the production of bacterial nanocellulose (BNC) from banana midrib juice (BMJ). The content is well constructed and can be considered for publication after addressing the following concerns.

1.     The authors should provide a more careful explanation of the reliability of their statistical model, for instance, equation (3).

2.     The authors should provide more experimental evidence that proves the highest BNC yield on the optimal point in Figure 2.

3.     How did the quality of BMJ-fermented BNC compared to those from regular H.S. culture medium?

4.     Following the previous question, the authors showcased the FTIR, TGA, and XRD data of their BNC. However, there is no comparison with benchmark groups, for instance, H.S. medium-cultivated BNC.

Author Response

(The authors gave the same response as above.)

Round 2

Reviewer 1 Report

Comments and Suggestions for Authors

Overall, the article shows significant improvements in the presentation of the data, but the analysis needs to be enhanced. Below, I have some additional questions for the authors:

Comment 4: There was no response to the requested question. ¿Were the ATR spectra mathematically processed for comparison with spectra obtained through transmission?

Figure 5: ¿Why does the ATR-FTIR of cellulose produced with the HS medium exhibit a broad peak in the OH region, while the sample obtained using the optimized SCD medium does not show this?

Figure 5: ¿What is the reason for the presence of absorbance in the range between 2000-250 cm-1 in the BC sample produced by the HS medium?

Figure 6: ¿Why does the HS sample have a lower residue in the TGA compared to the SCD sample?

Lines 460-471: ¿Why are the fibers produced in the SCD medium larger than those produced in the HS medium?

Line 481: What differences can be found between the ATR-FTIR spectra of bacterial cellulose samples produced in the SCD and HS mediums?

The explanations regarding the characterization results still lack arguments; they remain overly descriptive and fail to explain the reasons behind the differences observed in the two bacterial cellulose samples. The authors should improve these analyses for the article to be publishable."

Author Response

We deeply appreciate your detailed observations. Each of your comments has been thoughtfully considered, and we have composed a comprehensive, point-by-point response. We invite you to review our detailed answers in the attached document. Once again, we express our sincere gratitude for dedicating your time and effort to reviewing our document.

Round 3

Reviewer 1 Report

Comments and Suggestions for Authors

Line 481. The authors use a bibliographic reference to indicate that other researchers have explained the changes that occur when using a non-commercial substrate. However, it is crucial that they provide a technical explanation of what occurred, rather than just referencing it. It is necessary to technically elucidate the results of this research, building upon previous findings.

The references cited in the text are not found in the file provided. Please verify this.

Comments on the Quality of English Language

The text, overall, requires minor corrections in English.

Author Response

(The authors gave the same response as above.)
